biophysics

phase transitions, population collapse, *Saccharomyces cerevisiae*, *Escherichia coli*, temperature stress, salt stress

**Author for correspondence:**
Sonya Bahar
e-mail: bahars@umsl.edu

# Phase transition behaviour in yeast and bacterial populations under stress

Stephen W. Ordway[1], Dawn M. King[1], David Friend[1,2], Christine Noto[2], Snowlee Phu[2], Holly Huelskamp[2], R. Fredrik Inglis[2], Wendy Olivas[2] and Sonya Bahar[1]

[1]Department of Physics and Astronomy, and [2]Department of Biology, University of Missouri – St. Louis, Saint Louis, MO, USA

(iD) DMK, 0000-0002-3665-3746; SB, 0000-0002-9064-9377

Non-equilibrium phase transitions from survival to extinction have recently been observed in computational models of evolutionary dynamics. Dynamical signatures predictive of population collapse have been observed in yeast populations under stress. We experimentally investigate the population response of the budding yeast *Saccharomyces cerevisiae* to biological stressors (temperature and salt concentration) in order to investigate the system's behaviour in the vicinity of population collapse. While both conditions lead to population decline, the dynamical characteristics of the population response differ significantly depending on the stressor. Under temperature stress, the population undergoes a sharp change with significant fluctuations within a critical temperature range, indicative of a continuous absorbing phase transition. In the case of salt stress, the response is more gradual. A similar range of response is observed with the application of various antibiotics to *Escherichia coli*, with a variety of patterns of decreased growth in response to antibiotic stress both within and across antibiotic classes and mechanisms of action. These findings have implications for the identification of critical tipping points for populations under environmental stress.

## 1. Introduction

As our planet edges ever closer to the 'Sixth Extinction' [1–4], environmental stressors are increasing, and the ecological effects of these stressors on ecosystems are becoming more apparent. As a result, identification of early warning signs of population collapse has become a priority [5–11]. Population decline has been suggested to be accompanied by critical slowing down [5–7,10], a dynamical phenomenon associated with scale-free, power-law dynamics, though other studies have observed population decline

in the absence of such critical signatures [12]. Importantly, due to critical slowing down, populations recover more slowly from perturbations in the neighbourhood of tipping points between survival and collapse [10]. The identification of indicators of incipient population decline is of increasing urgency in the light of collapse of pollinator [13–15] and avian [16] communities, among others. Recent studies have identified evidence for worldwide decline in insect populations [17–20], which could have catastrophic cascading effects on the global ecosystem.

Physics-based models of *phase transitions* provide useful models for extinction processes. In a phase transition, a system undergoes a change in the value of an *order parameter* characterizing the system's state, as a *control parameter* is varied. At a critical value of the control parameter, the system passes a tipping point and undergoes a change of state, such as from survival to extinction. The system's characteristic fluctuations in the neighbourhood of the transition can serve as a warning sign for the incipient state change. In *critical* (also called *continuous* or *second-order*) transitions, these fluctuations are characterized by a large standard deviation in the order parameter in the vicinity of the transition. The dynamics of the fluctuations can be quantified by *critical exponents*, which define the *universality class* of the phase transition, and hence the system's behaviour as it passes through the transition. Widely different systems can undergo phase transitions in the same universality class. In *non-equilibrium* phase transitions, a system undergoes a transition to an 'absorbing state', from which it cannot recover [21–22].

Computational evolutionary models of transitions from survival to extinction can be characterized as non-equilibrium phase transitions. A recent agent-based computational evolutionary model has been shown to exhibit a non-equilibrium phase transition with behaviour similar, but not identical, to the directed percolation universality class [23–27]. This behaviour has been observed during simulated transitions from survival to extinction as maximum mutation size [23–27] and death rate [26,27] are varied as control parameters.

Dai and colleagues have performed population dynamics experiments to investigate the stability and resilience of yeast cultures subjected to environmental stressors [8–11]. Studying their system through the lens of nonlinear dynamics, they mapped the dynamics of stable and unstable populations in response to environmental stressors such as dilution factor (a proxy for death rate) [8–10], nutrient (sucrose) concentration (a proxy for carrying capacity) [10] and osmotic stress (NaCl concentration) [8,10]. These studies resulted in a means of characterizing a population's resilience, i.e. its ability to recover from a large environmental perturbation [10].

In the present work, we adapt the experimental approach used by Dai and colleagues in order to investigate the decline of the budding yeast *Saccharomyces cerevisiae* and *Escherichia coli* populations under stress. We investigate the dynamics of yeast population decay in the presence of two different stressors, temperature and salt (NaCl) concentration, and the decay of *E. coli* populations in response to a range of antibiotics.

# 2. Material and methods

## 2.1. *Saccharomyces cerevisiae* experiments

Experiments were performed using the yWO3 [28] strain of *S. cerevisiae*. This strain was selected because it is a well-described wild-type laboratory strain that is neither thermophilic nor thermotolerant. To investigate the dynamics of yeast population growth under stress, two environmental stressors were used, temperature and elevated NaCl concentration. In both cases, *S. cerevisiae* was initially grown at 30°C, which is the optimum temperature for *S. cerevisiae* growth [29], in standard medium (YEPD) containing $10 \, \text{g} \, \text{l}^{-1}$ yeast extract, $20 \, \text{g} \, \text{l}^{-1}$ peptone and 2% dextrose. These initial 50 ml cultures in liquid media were inoculated from a plate with cells to an optical density (OD) of 0.0001 to allow multiple doublings in log phase over 24 h, thereby creating a large concentration of cells without saturating the culture. From the starting OD 0.0001 concentration, the cells typically grew to an OD of 2.5 in 24 h. These initial cultures were then used as starter cultures to inoculate fresh media for the temperature stress and salt stress experimental cultures. All OD measurements were taken at 600 nm and measured using a Turner visible spectrophotometer.

### 2.1.1. Temperature stress

Wild-type *S. cerevisiae* is known to grow optimally at 30°C [29] and to exhibit a sharp decline in growth rate at temperatures exceeding approximately 40°C [30]. To study the response to temperature stress, the

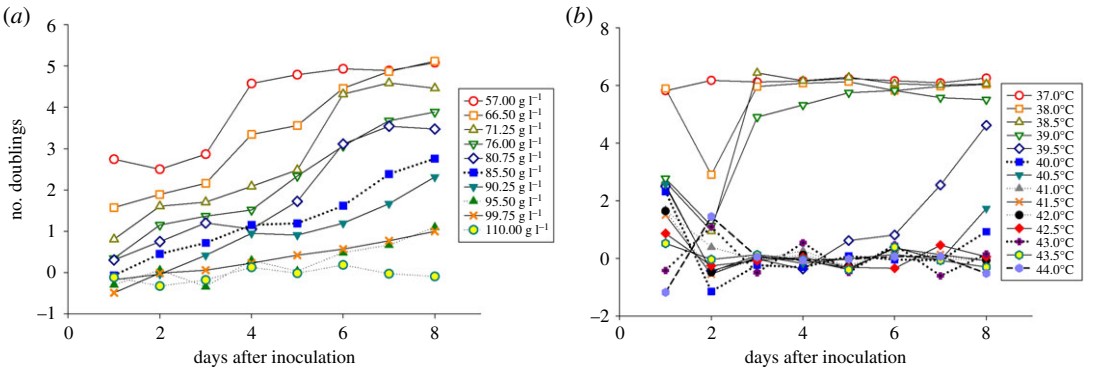

**Figure 1.** Number of *S. cerevisiae* population doublings is shown as a function of days after initial inoculation for various values of NaCl concentration (*a*) and temperature (*b*). $N \geq 3$ for each data point; error bars are not shown to avoid crowding the figure (see electronic supplementary material, table).

initial culture, prepared as described above, was inoculated into 50 ml of YEPD in 250 ml flasks to a resulting OD of 0.05. The samples were then placed in orbital shaking water baths that had already reached the particular temperature of interest for a given set of measurements. Sample ODs were then measured every 24 h. If the sample had grown, it was diluted back to an OD of 0.05 in a volume of 50 ml YEPD in a new 250 ml flask. If the sample had not grown significantly (less than 0.01 OD growth), or if the OD of the sample was lower than 0.05, the sample was placed back into the water bath for another 24 h. Measurements were taken over eight 24 h periods. Growth was measured by calculating the number of times the population doubled between measurement cycles, or number of doublings, using

$$n = \frac{(\log(\mathrm{OD}_{\mathrm{final}}/\mathrm{OD}_{\mathrm{initial}}))}{\log(2)}, \tag{2.1}$$

where $\mathrm{OD}_{\mathrm{initial}}$ is the OD at the beginning of each 24 h growth period, and $\mathrm{OD}_{\mathrm{final}}$ is the measured OD after each 24 h growth period.

For the measurements of growth under temperature stress, sample cultures were measured from 38°C to 44°C at 0.5°C intervals with a sample size $N \geq 3$, with an additional set of $N = 3$ measurements at 37°C. Since 37°C was well into the survival regime, as can be seen from the steady-state growth curve at this temperature in figure 1, and far from any population tipping points, measurements under this condition served as a reference point to confirm any emerging trends. Data points taken at 30°C (not shown) confirmed that the growth rate at this standard temperature for yeast growth was similar to the growth observed at 37°C.

### 2.1.2. NaCl stress

*Saccharomyces cerevisiae* growth was investigated over a range of salt concentrations, from concentrations under which cells are documented to grow normally to concentrations which are known to cause considerable stress, including cell death [31–35]. For the measurement of yeast growth under high salt stress, NaCl was added to freshly prepared YEPD in order to achieve the desired NaCl concentration. Culture growth was measured for NaCl concentrations from 66.5 to 104.5 g $\mathrm{l}^{-1}$ at approximately 5 g $\mathrm{l}^{-1}$ intervals with a sample size $N \geq 3$. An additional set of $N = 3$ measurements at 57 g $\mathrm{l}^{-1}$, which is well into the survival regime, as can be seen from the steady-state growth curve at this salt concentration in figure 1, served as a reference point to confirm any emerging trends.

An initial culture in YEPD was inoculated into 50 ml of YEPD + NaCl in 125 ml flasks to a resulting OD of 0.05. The samples were then placed in an orbital shaking water bath at 30°C. As with the temperature studies described above, sample ODs were measured after 24 h at 600 nm using a Turner visible spectrophotometer. If growth had occurred, the sample was diluted back to an OD of 0.05 in a volume of 50 ml YEPD + NaCl in a new 125 ml flask. If the sample did not have significant growth (less than 0.01 OD), or if the OD was less than 0.05, the sample was returned to the incubator until the next measurement period. Measurements were taken over eight 24 h periods, and doublings were calculated using equation (2.1). All data, for both temperature and salt stress, are available in [36].

**Table 1.** Mechanisms of action of antibiotics used.

| antibiotic | class | mechanism | mode of action |
|---|---|---|---|
| ampicillin | β-lactam | inhibits cell wall synthesis | bactericidal |
| carbenicillin | β-lactam | inhibits cell wall synthesis | bactericidal |
| chloramphenicol | amphenicol | protein synthesis (50S inhibitor) | bacteriostatic |
| ciprofloxacin | fluoroquinolone | inhibits DNA gyrase | bactericidal |
| gentamycin | aminoglycoside | protein synthesis (30S inhibitor) | bactericidal |
| kanamycin | aminoglycoside | protein synthesis (30S inhibitor) | bactericidal |
| rifampicin | rifamycin | DNA-directed RNA polymerase | bactericidal |
| spectinomycin | aminoglycoside | protein synthesis (30S inhibitor) | bacteriostatic |
| streptomycin | aminoglycoside | protein synthesis (30S inhibitor) | bactericidal |
| tetracycline | tetracycline | protein synthesis (30S inhibitor) | bacteriostatic |

## 2.2. *Escherichia coli* experiments

*Escherichia coli* (MG1655) were grown in M9 minimal glucose media overnight, shaking at 37°C. This bacterial culture was used to inoculate 96-well plates containing a dilution series of 10 different antibiotics, at a starting OD (600 nm) of 0.005 (i.e. 2 µl of culture in 198 µl media). We selected a panel of 10 antibiotics across different antibiotic classes with a variety of mechanisms of action (table 1). Each antibiotic was added at a starting concentration of 100 µg ml$^{-1}$ and subsequently serially diluted by ¾ to give 40 different concentrations, ranging from 100 to 0.001 µg ml$^{-1}$. There were six replicates for each antibiotic. The 96-well plates were incubated, shaking at 37°C, for 24 h. The OD (600 nm) for each well was measured using a Cytation 3 multimode plate reader (BioTek). Doubling times for each well were calculated as a function of initial and final OD using equation (2.1) as described above. All data are available in [36].

## 3. Results

As shown in figure 1, increases in both environmental stressors caused a decrease in yeast growth rate, as would be expected. However, the time course of the stress response differed significantly between the stressors. The data shown in figure 1a indicates that increasing salt concentrations caused the yeast to grow more slowly, but with a gradual shift until growth ceased altogether. By contrast, as shown in figure 1b, as the temperature was increased, a sharp drop in doubling rate was observed. Here, the cultures changed from a quick recovery time in the 37–39°C range, to a recovery after a long lag time (39.5–40.5°C range) and finally to being unable to recover at all for temperatures above 40.5°C. Standard deviations are provided in electronic supplementary material, Data tables 1 and 2, but not in figure 1, to avoid crowding the figures.

In order to investigate the population dynamics in the neighbourhood of the 'tipping point' into decline, average growth over the last 3 days was calculated as a function of NaCl concentration (figure 2a) and temperature (figure 2b). As can be seen in figure 1, in most cases, the populations had reached an approximately constant growth rate by post-inoculation days 6–8. Averaging the growth rates over these three days thus serves as a proxy for a 'steady-state' growth rate. In figure 2, these averages are shown, with a grand average taken over all experiments at each condition. This gives a measure of the system's overall response to each stressor.

It can be seen from figure 2 that, for both stressors, the yeast population ultimately experienced an environment too harsh for survival. However, in the case of increasing salt concentrations (figure 2a), the system underwent a smooth, gradual decline. Minimal fluctuations were observed in the transition region, and there was no identifiable tipping point. In stark contrast, the response to temperature stress (figure 2b) showed an abrupt drop in growth rate, with comparatively large fluctuations in the transition region, indicating a tipping point around 39.5°C.

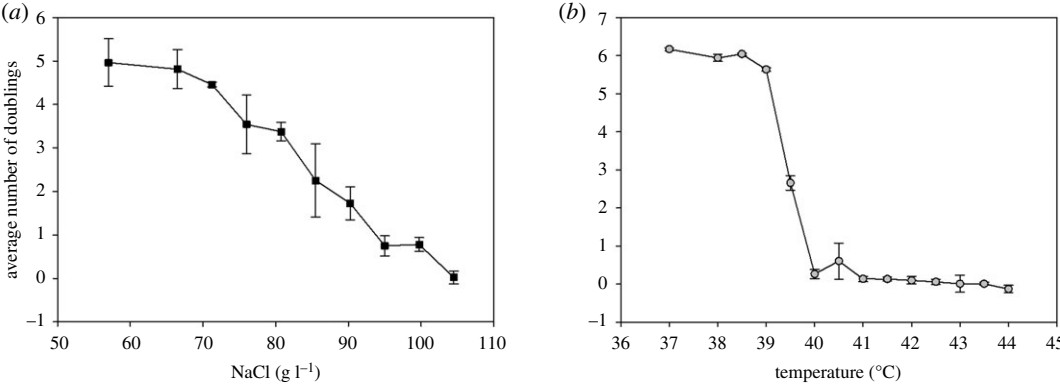

**Figure 2.** Steady states of the *S. cerevisiae* population for NaCl stress (*a*) and temperature stress (*b*) are shown. For each condition, the populations are first averaged over days 6–8 post-inoculation, then over all runs at that condition and finally averaged over all experiments at that condition. Error bars show standard deviation.

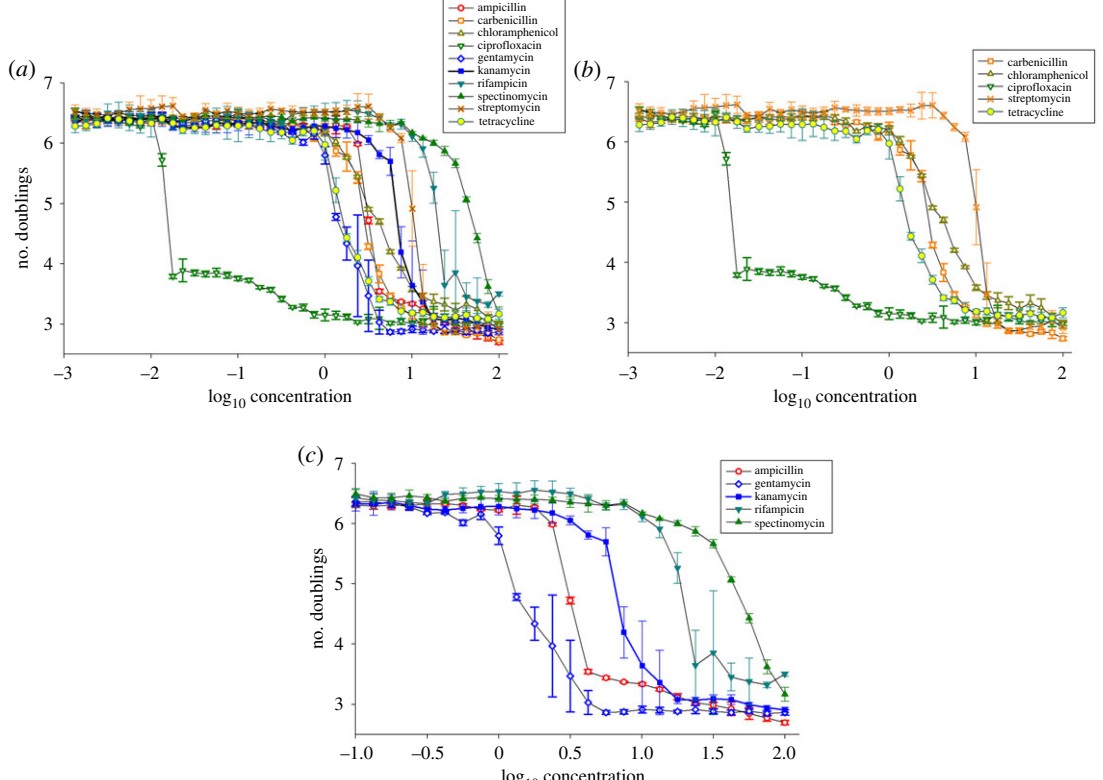

**Figure 3.** (*a*) Number of doublings of *E. coli* (MG1655 WT) populations as a function of $\log_{10}$(antibiotic concentration) for ten antibiotics. Error bars show standard deviation over six replicates for each antibiotic concentration. (*b*) Number of doublings for five of the studied antibiotics over a smaller concentration range, in order to show detail. (*c*) Number of doublings for the other five studied antibiotics over a smaller concentration range, in order to show detail.

In figure 3, we show the decay curves of *E. coli* populations in the presence of various antibiotics. All the antibiotics tested are shown in figure 3*a*. In figure 3*b* and *c*, the responses to five each of the ten antibiotics studied are shown (on a shorter horizontal scale, in the case of 3*c*), in order to display the details of the responses more clearly. The presence of antibiotic stress caused a decrease in bacterial growth across all the antibiotics tested, as expected. However, we observed a variety of patterns of decreased growth in response to antibiotic stress both within and across antibiotic classes and mechanisms of action. This ranged from a very sharp drop in bacterial growth in the presence of ciprofloxacin (open green triangles, figure 3*a* and *b*) to more gradual declines in the presence of spectinomycin (filled green triangles, figure 3*a* and *c*). This broad range of response patterns is qualitatively similar to those observed in the yeast experiments.

# 4. Discussion

We have examined the response of S. cerevisiae to two different environmental stressors, temperature and NaCl concentration, as well as the response of E. coli to various antibiotics. In contrast with the studies of Dai et al. [8–10], our yeast experiments used glucose as a nutrient source rather than sucrose. Glucose can be directly metabolized by S. cerevisiae, in contrast with the disaccharide sucrose, which must be hydrolysed outside the cytoplasm. However, most of the hydrolysis products of sucrose diffuse away before they can be taken back up into the cytoplasm, becoming public goods [37]. As a result, yeast cells fed with sucrose exhibit a classical Allee effect, with maximal growth at an intermediate population density [8]. Use of glucose rather than sucrose prevents population density from having a significant effect on survival, enabling temperature and salt concentration to serve as isolated stress variables in the studies described here.

We find that, while both high temperature and high NaCl concentration have a negative effect on the yeast doubling rate, the dynamics of the response to the two stressors is significantly different. In particular, the response to NaCl is gradual (figure 2a), while temperature stress induces a sharp drop reminiscent of a critical phase transition with a well-defined 'tipping point' (figure 2b). Accompanying this change in population growth, the fluctuations of the system are much higher in the vicinity of the tipping point as the temperature increases. In the E. coli experiments (figure 3), the responses to some of the antibiotics, such as ciprofloxacin, streptomycin and kanamycin, exhibited critical phase-transition-like behaviour, while others, such as chloramphenicol and spectinomycin, produced a gradual response without a well-defined tipping point. For other antibiotics, the response was less clearly defined.

In order to observe any critical behaviour, it is necessary to examine the system after it has stabilized. As figure 1 shows, the yeast had fluctuating growth behaviour in the first few days after exposure to both stressors. For most control parameter values, the behaviour stabilized after a few days. It is this stable 'long-term' behaviour that is averaged and shown in figure 2. Our results are consistent with those of Mensonides et al., who demonstrated the initial slowing of S. cerevisiae population growth due to temperature for budding yeast, accompanied by changes in metabolism, by observing the populations for 6 h after introduction to various temperatures [30]. They 'observed a surprisingly "thin line" for cells between growing, surviving and dying, with regard to growth temperature' [30]. The sharp change between survival and death observed by Mensonides and colleagues occurred for a slightly higher temperature (between 42 and 43°C) than that observed in the present paper (39.5°C). This difference can be explained as the result of short- versus long-term behaviour; in all our cultures below 42°C, there was some measurable amount of growth in the first 24 h period, but the populations were unable to maintain consistent growth above 39.5°C, as can be seen in figure 1a. Mensonides et al. [30] suggested that 'it cannot be concluded if [the] effect on growth and viability is caused by the absolute temperature, or by the difference between initial and new temperature'. Notably, the initial temperature in their study was 28°C, while it was 30°C in our experiments. Dai et al. [10] also observed that S. cerevisiae populations lose stability at different rates and exhibit different levels of resilience, for different environmental stressors [10]. Specifically, for S. cerevisiae grown in the presence of sucrose, populations lost stability more rapidly (and early warning signals based on stability loss performed better) when stressed by increasing dilution factor, which serves as a proxy for death rate in S. cerevisiae populations grown on sucrose, than in response to simple nutrient depletion [10].

Why is the response to temperature stress much more abrupt than the salt stress response? Mensonides et al. proposed that the 'narrow temperature range' over which population decline is observed 'may be explained by assuming the rapid denaturation of one or a very limited number of enzymes which are essential to growth' [10]. This is consistent with the current understanding of heat shock response [38] and more specifically of protein denaturation [39] in S. cerevisiae. By contrast, these cells have a variety of responses that can modulate the response to salt stress. Changes in salt concentration can have multiple effects on yeast, including osmotic shock [40], direct toxicity from $Na^+$ ions [31] and changes in membrane potential [37]. Saccharomyces cerevisiae can respond to these diverse insults with a range of responses, such as activation of the high-osmolarity glycerol (HOG) pathway [39] and membrane depolarization accompanied by decreased permeability [32]. After observing an increase in abundance of a dozen plasma membrane proteins in response to mild salt stress, Szopinska et al. [33] proposed that protein internalization occurs rapidly after hyper-osmotic or ionic shock, enabling a cell to remain viable until a slower transcriptional response [34,35] can be activated. These various studies suggest a model in which S. cerevisiae is better able to modulate its response to salt stress than to heat stress, consistent with the more gradual response to salt stress we observe.

It is less clear, however, why *E. coli* responds so differently to antibiotics. Many of the antibiotics tested have the same mode of action but display completely different decay curves (see, for example, spectinomycin and kanamycin, which are both aminoglycosides). *A priori* one might expect that bacteriostatic antibiotics that only inhibit growth, as opposed to bacteriocidal antibiotics that kill *E. coli*, might have more gradual decay curves. However, there was a large variation in decay curves both within bacteriostatic (e.g. chloramphenicol and tetracycline) and bacteriocidal (e.g. rifampicin and ampicillin) antibiotics. It may be that the inherent pharmacokinetics of each antibiotic drives these differences in bacterial killing, and more comprehensive screening would be required to elucidate the exact underlying mechanisms. The differences observed in these bacterial decay curves might have important implications for the evolution of antibiotic resistance. It is possible that mutations could shift the shape of the decay curve; this could be tested in experimental evolution studies.

The system behaviour seen in the temperature stress experiments and some types of antibiotics (e.g. ciprofloxacin, streptomycin, kanamycin) match exceptionally well with critical phase transition behaviour, while the smooth transition in response to NaCl stress and other antibiotics (e.g. spectinomycin and chloramphenicol) show none of the signifying characteristics of a critical phase transition. These findings demonstrate that monitoring the dynamics of population decay can not only provide a warning sign of incipient collapse but could also be used in order to identify the type of stressor that is causing the destabilization. The difference in these two behaviours demonstrates that, without reference to the environmental stressors, simple observation of a population in decline may be insufficient to predict the course of the population's progress toward collapse. With detailed information about individual stressor effects, however, it may be possible to identify the driving stressor(s) of a population decline and determine whether it is approaching a critical collapse, or a gradual decline based on the transition behaviour. Understanding the type of stress response could be critical in designing appropriate intervention protocols.

The type of stress response may also play a role in the dynamics of population recovery. This has been observed by King *et al.* in a computational evolutionary model when a system is near the tipping point in a critical phase transition [26,27] and in the studies of population resilience in the presence of different stressors by Dai *et al.* [8–10]. As mentioned above, a key behaviour in systems with critical transition behaviour is critical slowing down. If intervention is taken to save a population from collapse, systems exhibiting critical slowing down will have significantly longer recovery times compared to systems undergoing gradual transitions. These systems will require more stringent monitoring to ensure that recovery continues through the long recovery period; it may also take far longer for significant signs of recovery to appear. Identification of transition type can be crucial in resource allocation to rescue the highest number of populations approaching collapse and prevent premature abandonment of a recovering population. Lastly, it is important to note that, due to cooperative effects, population size itself may modulate the dynamics of collapse and recovery [41] and should be taken into account when designing such interventions.

# 5. Conclusion

We observe population declines exhibiting characteristics of a critical phase transition in both yeast (*S. cerevisiae*) and bacteria (*E. coli*). In the case of yeast, phase transition behaviour is only observed when cultures are subjected to temperature stress, but not in the case of high salt concentration. In the latter case, the yeast populations exhibit a gradual decline without any sign of criticality. Similarly, in bacteria, some antibiotics produce population declines characteristic of a critical phase transition, while others produce more gradual declines. Taken together, the yeast and bacterial results suggest that phase-transition-like population dynamics may occur in response to a broad range of stressors in different organisms. Further, the disparity between phase-transition-like behaviour and more gradual population declines in both eukaryotic and prokaryotic cells for different stressors raises the possibility that divergent population dynamics may also occur for a range of organisms and stressors.

With increasing peril of current climate change and the approach of 'the Sixth Extinction', it is crucial to examine population dynamics for early indicators of collapse. Investigating population collapse through the lens of critical phase transitions can identify factors that drive population dynamics in uncontrolled environments and can facilitate the development of protocols for population rescue. This approach may also be used to pinpoint what stressor or combination of stressors is responsible for pushing a population to collapse, as well as to design intervention and rescue protocols.

Our results indicate that critical phase transition behaviour occurs in the presence of some stressors but not others. How and why this occurs is an avenue for future investigations, which could include, among other avenues, investigation of metabolic flux and imaging analysis of cells in the presence of different stressors over time. In the case of temperature and salt stress, recent experimental evolution studies in yeast [42–45], as well as the extensive literature on transcriptional response to stress in *S. cerevisiae* [35], could point the way toward identifying particular genetic and/or transcriptional determinants of population collapse dynamics. The repressors Nrg1 and Nrg2 have already been implicated in *S. cerevisiae* adaptation to salt stress [46]. Nearly two decades ago, a group of several hundred yeast genes were shown to undergo significant changes in expression in response to environmental stress [47,48]. The genetic pathways that mediate resilience are likely to be complicated, multilayered and historically contingent. Berry *et al.* [49] found that the changes in gene expression that confer tolerance to $H_2O_2$ stress in yeast differ depending on the prior mild insult the cells received (salt, heat shock or dithiothreitol (DTT)). While such complex interactions make investigating such problems daunting, they also offer the tantalizing possibility that mapping gene activation can reveal a cell's environmental history, much as a human patient's complement of antibodies can show their history of disease exposure. Such studies could ultimately contribute not only to the development of population rescue protocols but may also facilitate the genetic engineering of species more resilient in the face of environmental stress.

Data accessibility. Data are available at the Dryad Digital Repository, https://doi.org/10.5061/dryad.5hqbzkh2k [36].

Authors' contributions. S.W.O. carried out yeast experiments and data analysis, and participated in drafting the manuscript; D.M.K. designed the yeast study, performed preliminary yeast experiments and participated in drafting the manuscript; D.F., C.N. and S.P. performed yeast experiments and data analysis; H.H. performed *E. coli* experiments and data analysis; R.F.I. designed the *E. coli* experiments and participated in drafting the manuscript; W.O. designed the yeast study and participated in drafting the manuscript; S.B. designed the study, performed data analysis and participated in drafting the manuscript. All authors gave approval for publication.

Competing interests. The authors have no competing interests to declare.

Funding. S.W.O., D.M.K., S.B. and W.O. were supported by a University of Missouri Interdisciplinary Intercampus Research Grant.

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
