## [Reviewer comments · Royal Society Open Science]

Review History

RSOS-192211.R0 (Original submission)

Review form: Reviewer 1

Is the manuscript scientifically sound in its present form?

No

Are the interpretations and conclusions justified by the results?

Yes

Is the language acceptable?

Yes

Do you have any ethical concerns with this paper?

No

Have you any concerns about statistical analyses in this paper?

No

Recommendation?

Accept with minor revision (please list in comments)

Comments to the Author(s)

Major comments

1. The Material and Method section is not well detailed. Some procedures are unclear and the parameters chosen for stressors are missing, such as the temperatures and the concentrations of NaCl tested. Please reorder this section with all the details necessary to a good comprehension and reproducibility of the procedures employed.

2. Authors considered 37°C and 57 g/L salt concentration as reference points to confirm any emerging trends during growth under stressing conditions. Based on what evidence did the authors make this choice? and why did they not include as control, or additional control, the growth of the culture without stressing compounds? Please motivate.

3. Why did the authors chose the *S. cerevisiae* yWO3 strain for this type of experiment? Is a thermotolerant or a thermophilic strain?

Page 5, lines 8-13: It is not clear on the basis of what evidence authors concluded that bacteria patters are applicable to a broad range of stressors in different organisms. Please explain and rephrase.

Minor comments

Page 4, line 26: Please move the reference eleven into the brackets of references 8-10.

Page 5, line 29: Why did the authors chose 30°C as temperature for initial growth of *S. cerevisiae*? Please explain.

Page 5, lines 31-36. Please clarify why the pre-cultures has been inoculated at such a low OD600 0.0001. From plates or from pre-precultures in liquid medium? Please specify the instrument used for the measurement of OD600 nm.

Page 6, line 18. ".....separate YEPD + NaCl media were prepared for each NaCl concentration". This procedure is not clear. What is meant by NaCl culture medium? Or did the authors prepare a stock solution of NaCl at a specific concentration and then obtain the concentrations tested in YEPD by dilution? Please clarify.

Page 6, line 39. ".....All data is available in [29]". I invite the authors to indicate data repository not as a reference.

Page 7, lines 17-27. Please move these two sentences into the M&M section. These details belong to the experimental design and is more appropriate to insert them into that specific section. Please motivate the choice of the range of temperatures and salt concentrations tested.

Page 9, line 54. "....., streptomycin,". Please remove the comma before and.

Page 10, line 15. Please correct "application of both stressors with "exposition to both stressors".

Review form: Reviewer 2

Is the manuscript scientifically sound in its present form?

No

Are the interpretations and conclusions justified by the results?

Yes

Is the language acceptable?

Yes

Do you have any ethical concerns with this paper?

No

Have you any concerns about statistical analyses in this paper?

No

Recommendation?

Reject

Comments to the Author(s)

The manuscript is trying to find out critical time of transition among the population of yeast and bacteria (*E. coli*) under environmental stressors (temperature, antibiotics and salt concentration).

Data presented in the manuscript are too preliminary and far from drawing any conclusion. I would expect to see what metabolic flux is happening under different stressors in order to validate the data. Or they would done some microbiological imaging analyses to see differences in cell shaping under those stressors. Overall the manuscript doesn't meet the standard of the journal.

In minor:

Introduction: it is consisted of relevant information, however, conclusion from the introduction should be avoided.

Line 45, page 6: What was the temperature of interest? Similarly, what were concentration of NaCl?

Decision letter (RSOS-192211.R0)

02-Mar-2020

Dear Dr Bahar,

The editors assigned to your paper ("Phase Transitions in Yeast and Bacterial Populations Under Stress") have now received comments from reviewers. We would like you to revise your paper in accordance with the referee and Associate Editor suggestions which can be found below (not including confidential reports to the Editor). Please note this decision does not guarantee eventual acceptance.

Please submit a copy of your revised paper before 25-Mar-2020. Please note that the revision deadline will expire at 00.00am on this date. If we do not hear from you within this time then it will be assumed that the paper has been withdrawn. In exceptional circumstances, extensions may be possible if agreed with the Editorial Office in advance. We do not allow multiple rounds of revision so we urge you to make every effort to fully address all of the comments at this stage. If deemed necessary by the Editors, your manuscript will be sent back to one or more of the original reviewers for assessment. If the original reviewers are not available, we may invite new reviewers.

- Data accessibility

If you wish to submit your supporting data or code to Dryad (<http://datadryad.org/>), or modify your current submission to dryad, please use the following link:
<http://datadryad.org/submit?journalID=RSOS&manu=RSOS-192211>

- Competing interests

- Authors' contributions

- Acknowledgements

- Funding statement

on behalf of Dr Ulas Tezel (Associate Editor) and Pietro Cicuta (Subject Editor)
openscience@royalsociety.org

Reviewers' Comments to Author:
Reviewer: 1

Comments to the Author(s)

Major comments

1. The Material and Method section is not well detailed. Some procedures are unclear and the parameters chosen for stressors are missing, such as the temperatures and the concentrations of NaCl tested. Please reorder this section with all the details necessary to a good comprehension and reproducibility of the procedures employed.

2. Authors considered 37°C and 57 g/L salt concentration as reference points to confirm any emerging trends during growth under stressing conditions. Based on what evidence did the authors make this choice? and why did they not include as control, or additional control, the growth of the culture without stressing compounds? Please motivate.

3. Why did the authors chose the *S. cerevisiae* yWO3 strain for this type of experiment? Is a thermotolerant or a thermophilic strain?

Page 5, lines 8-13: It is not clear on the basis of what evidence authors concluded that bacteria patters are applicable to a broad range of stressors in different organisms. Please explain and rephrase.

Minor comments

Page 4, line 26: Please move the reference eleven into the brackets of references 8-10.

Page 5, line 29: Why did the authors chose 30°C as temperature for initial growth of *S. cerevisiae*? Please explain.

Page 5, lines 31-36. Please clarify why the pre-cultures has been inoculated at such a low OD600 0.0001. From plates or from pre-precultures in liquid medium? Please specify the instrument used for the measurement of OD600 nm.

Page 6, line 18. ".....separate YEPD + NaCl media were prepared for each NaCl concentration". This procedure is not clear. What is meant by NaCl culture medium? Or did the authors prepare a stock solution of NaCl at a specific concentration and then obtain the concentrations tested in YEPD by dilution? Please clarify.

Page 6, line 39. ".....All data is available in [29]". I invite the authors to indicate data repository not as a reference.

Page 7, lines 17-27. Please move these two sentences into the M&M section. These details belong to the experimental design and is more appropriate to insert them into that specific section.

Please motivate the choice of the range of temperatures and salt concentrations tested.

Page 9, line 54. "....., streptomycin,". Please remove the comma before and.

Page 10, line 15. Please correct "application of both stressors with "exposition to both stressors".

Reviewer: 2

Comments to the Author(s)

The manuscript is trying to find out critical time of transition among the population of yeast and bacteria (*E. coli*) under environmental stressors (temperature, antibiotics and salt concentration).

Data presented in the manuscript are too preliminary and far from drawing any conclusion. I would expect to see what metabolic flux is happening under different stressors in order to validate the data. Or they would done some microbiological imaging analyses to see differences in cell shaping under those stressors. Overall the manuscript doesn't meet the standard of the journal.

In minor:

Introduction: it is consisted of relevant information, however, conclusion from the introduction should be avoided.

Line 45, page 6: What was the temperature of interest? Similarly, what were concentration of NaCl?

Author's Response to Decision Letter for (RSOS-192211.R0)

See Appendix A.

Decision letter (RSOS-192211.R1)

Dear Dr Bahar,

It is a pleasure to accept your manuscript entitled "Phase Transitions in Yeast and Bacterial Populations Under Stress" in its current form for publication in Royal Society Open Science.

on behalf of Dr Ulas Tezel (Associate Editor) and Pietro Cicuta (Subject Editor)
openscience@royalsociety.org

Appendix A

Response to Referees

We thank the referees very much for their useful comments. We have revised the manuscript in response to their comments, and detail our responses and revisions below in **red**.

Reviewers' Comments to Authors:

Reviewer 1

Major comments

1. The Materials and Method section is not well detailed. Some procedures are unclear and the parameters chosen for stressors are missing, such as the temperatures and the concentrations of NaCl tested. Please reorder this section with all the details necessary to a good comprehension and reproducibility of the procedures employed.

Some of this information was included in at the beginning of the Results section rather than in Materials and Methods. We have moved this text into the Materials and Methods section for clarity.

2. Authors considered 37°C and 57 g/L salt concentration as reference points to confirm any emerging trends during growth under stressing conditions. Based on what evidence did the authors make this choice? and why did they not include as control, or additional control, the growth of the culture without stressing compounds? Please motivate.

These conditions are non-stress conditions, based on the range of temperatures and salt concentrations tolerated by wild-type *S. cerevisiae*. This can also be seen from the time series plots shown in Figure 1, which reach steady-state growth plateaus at these values. We have clarified this in the revised manuscript.

3. Why did the authors choose the *S. cerevisiae* yWO3 strain for this type of experiment? Is a thermotolerant or a thermophilic strain?

This strain was chosen because it is a well-described wild type laboratory strain. It is neither thermophilic nor thermotolerant. We have clarified this in the Materials and Methods section when we first describe the choice of this strain for our experiments.

Page 5, lines 8-13: It is not clear on the basis of what evidence authors concluded that bacteria patterns are applicable to a broad range of stressors in different organisms. Please explain and rephrase.

[Note that the text mentioned has been moved to the Conclusion following the suggestion of the second referee to remove “spoilers” from the Introduction.] We intended to make the point that the bacterial results, taken together with the yeast results, may suggest that phase-transition-like population dynamics occurs in response to a broad range of stressors in different organisms, and furthermore that the observation of the disparity between phase-transition-like behavior and more gradual population declines in both eukaryotic and prokaryotic cells suggests the possibility that this behavior may occur for a range of organisms and stressors. We have revised the text to make this clearer.

Minor comments

Page 4, line 26: Please move the reference eleven into the brackets of references 8-10.

Done.

Page 5, line 29: Why did the authors chose 30°C as temperature for initial growth of *S. cerevisiae*? Please explain.

30°C is the optimum temperature for yeast growth; we have clarified this in the text and added a reference.

Page 5, lines 31-36. Please clarify why the pre-cultures has been inoculated at such a low OD600 0.0001. From plates or from pre-cultures in liquid medium? Please specify the instrument used for the measurement of OD600 nm.

Pre-cultures in liquid medium were inoculated (from a plate) to OD600=0.0001 in order to ensure that the cells were in log phase without being saturated, so as to prevent the cells from entering the stationary phase. OD measurements were performed on a Turner Visible Spectrophotometer. The text has been revised to clarify these points.

Page 6, line 18. “.....separate YEPD + NaCl media were prepared for each NaCl concentration”. This procedure is not clear. What is meant by NaCl culture medium? Or did the authors prepare a stock solution of NaCl at a specific concentration and then obtain the concentrations tested in YEPD by dilution? Please clarify.

We intended to convey the fact that YEPD was freshly prepared for each NaCl concentration, rather than being taken from a single pre-made YEPD stock. We have revised the text to clarify this point.

Page 6, line 39. “.....All data is available in [29]”. I invite the authors to indicate data repository not as a reference.

Our understanding from the journal was that we are required to list the data repository as a reference. We were specifically instructed to “include a Data Availability section after your main text stating where supporting data and code are available; these details should be appropriately cited in both the reference list and data accessibility section of the manuscript.”

Page 7, lines 17-27. Please move these two sentences into the M&M section. These details belong to the experimental design and is more appropriate to insert them into that specific section.

Done.

Please motivate the choice of the range of temperatures and salt concentrations tested.

The ranges of temperature and salt concentrations used were selected because they started within a range that does not cause stress to *S. cerevisiae* populations, and extends into ranges that are well documented to cause stress and, therefore, population decline. This has been clarified both in the Materials and Methods section, and also in the Introduction and the Discussion.

Page 9, line 54. “....., streptomycin,”. Please remove the comma before and.

Done.

Page 10, line 15. Please correct “application of both stressors” with “exposition to both stressors”.

We have revised this to read “exposure to both stressors”.

Reviewer 2

Comments to the Author(s)

The manuscript is trying to find out critical time of transition among the population of yeast and bacteria (*E. coli*) under environmental stressors (temperature, antibiotics and salt concentration).

Data presented in the manuscript are too preliminary and far from drawing any conclusion. I would expect to see what metabolic flux is happening under different stressors in order to validate the data. Or they would done some microbiological imaging analyses to see differences in cell shaping under those stressors. Overall the manuscript doesn't meet the standard of the journal.

Our intent was to investigate population dynamics in response to various stressors in prokaryotic and eukaryotic systems. We would argue that we have clearly shown (1) that phase-transition-like dynamics clearly occurs in these two systems, and that (2) both systems exhibit a range of responses from phase-transition-like to gradual decline, depending on the stressor. We argue that this result has important implications, and therefore merits publication. Further experiments such as studies of metabolic flux or imaging analysis are

important next steps but are beyond the scope of the present work. We have added a mention metabolic flux and imaging analysis as important future directions in last paragraph of the Conclusions.

In minor:

Introduction: it is consisted of relevant information, however, conclusion from the introduction should be avoided.

The last paragraph of the Introduction has been revised and moved to the Conclusions.

Line 45, page 6: What was the temperature of interest? Similarly, what were concentration of NaCl?

The temperature of interest simply refers to the temperature at which the current data points were being taken. The text has been clarified to avoid confusion. The concentrations of NaCl (and the temperatures) were described at the beginning of the results section; this has been moved to the Materials and Methods section for clarity.